physiology, developmental biology

*Apis mellifera*, caste determination, diet, royal jelly

**Author for correspondence:**
Julia H. Bowsher
e-mail: julia.bowsher@ndsu.edu

# Diet quantity influences caste determination in honeybees (*Apis mellifera*)

Garett P. Slater[1], George D. Yocum[2] and Julia H. Bowsher[1]

[1]Department of Biological Sciences, North Dakota State University, PO Box 6050, Fargo, ND 58108, USA
[2]Biosciences Research Laboratory, USDA-ARS Edward T. Schafer Agricultural Research Center, 1605 Albrecht Boulevard, Fargo, ND 58102-2765, USA

 JHB, 0000-0001-6347-9635

In species that care for their young, provisioning has profound effects on offspring fitness. Provisioning is important in honeybees because nutritional cues determine whether a female becomes a reproductive queen or sterile worker. A qualitative difference between the larval diets of queens and workers is thought to drive this divergence; however, no single compound seems to be responsible. Diet quantity may have a role during honeybee caste determination yet has never been formally studied. Our goal was to determine the relative contributions of diet quantity and quality to queen development. Larvae were reared *in vitro* on nine diets varying in the amount of royal jelly and sugars, which were fed to larvae in eight different quantities. For the middle diet, an ad libitum quantity treatment was included. Once adults eclosed, the queenliness was determined using principal component analysis on seven morphological measurements. We found that larvae fed an ad libitum quantity of diet were indistinguishable from commercially reared queens, and that queenliness was independent of the proportion of protein and carbohydrate in the diet. Neither protein nor carbohydrate content had a significant influence on the first principle component 1 (PC1), which explained 64.4% of the difference between queens and workers. Instead, the total quantity of diet explained a significant amount of the variation in PC1. Large amounts of diet in the final instar were capable of inducing queen traits, contrary to the received wisdom that queen determination can only occur in the third instar. These results indicate that total diet quantity fed to larvae may regulate the difference between queen and worker castes in honeybees.

## 1. Introduction

Eusocial organisms have a division of labour in females between non-reproductive and reproductive individuals [1]. In bees, queens and workers have analogous genotypes; yet, these similar genomes produce distinct queen and worker phenotypes. Caste determination cues vary by species, but nutrition drives queen development in many social Hymenoptera [1,2]. In honeybees, nurse bees provision larvae with glandular secretions called jelly, which contain all the macronutrients and micronutrients required for growth and development [3]. It is thought that quality differences between royal jelly, which is fed to queen-destined larvae, and worker jelly controls this queen–worker dimorphism [2,3]; however, qualitative differences between royal and worker jelly have failed to fully explain caste determination. Recent findings suggest that diet quantity may have a significant role in honeybee caste determination [4,5]. However, the contribution of diet quantity has never been formally tested.

Since the 1890s, diet quality has been thought to determine caste in honeybees [6–10] through a 'biological active substance' found only in royal jelly

**Table 1.** Macronutrient content of diet treatments. (Each diet combination (i.e. high protein, high carbohydrates) was fed at every diet quantity (160–370 µl). The ad libitum treatment was fed the medium protein, medium-carbohydrate diet.)

| diet treatments | | ingredients | | | | | macronutrient (%) | | | |
|---|---|---|---|---|---|---|---|---|---|---|
| protein | carbohydrates | royal jelly (g) | glucose (g) | fructose (g) | yeast (g) | water (ml) | protein | carbohydrate | water | P : C ratio |
| high | high | 65 | 6 | 12 | 1 | 35 | 6.75 | 29.87 | 60.00 | 1 : 4.4 |
| | medium | 65 | 4 | 8 | 1 | 35 | 7.10 | 26.15 | 63.19 | 1 : 3.7 |
| | low | 65 | 2 | 4 | 1 | 35 | 7.50 | 22.01 | 66.73 | 1 : 2.9 |
| medium | high | 50 | 6 | 12 | 1 | 35 | 5.94 | 30.29 | 60.58 | 1 : 5.1 |
| | medium | 50 | 4 | 8 | 1 | 35 | 6.30 | 26.02 | 64.29 | 1 : 4.1 |
| | low | 50 | 2 | 4 | 1 | 35 | 6.71 | 21.20 | 68.48 | 1 : 3.2 |
| low | high | 35 | 6 | 12 | 1 | 35 | 4.86 | 30.84 | 61.35 | 1 : 6.3 |
| | medium | 35 | 4 | 8 | 1 | 35 | 5.21 | 25.84 | 65.78 | 1 : 5.0 |
| | low | 35 | 2 | 4 | 1 | 35 | 5.61 | 20.06 | 70.91 | 1 : 6 |

[6,11] that activates a binary developmental switch for queen development [12]. The quality hypothesis arose from early observations of the queen and worker larvae receiving different proportions of water-clear and milky-white secretions from nurse bee glands [3,9]. The secretion fed to queen-destined larvae was termed royal jelly [13], and was thought to contain the major dietary component driving queen development [6]. As a result of these early observations, nearly every major component in royal jelly has been tested for its effects on caste determination, with most studies finding positive results. Lipids [14], proteins [15,16], carbohydrates [17–19] and water [20] all contribute to queen development in honeybees under some experimental conditions. However, most of these studies did not control for the quantity of diet. Nurse bees provide queen-destined larvae with an excess of diet [3,6,8,21]. Diet quantity appears to have a significant, but unaccounted role in honeybee caste determination and may explain why so many qualitative aspects of diet have been linked with caste.

Diet quantity controls queen–worker phenotypes in many other social Hymenoptera [22–26]. Increased provisioning drives queen differentiation in cape honeybees [22], bumblebees [24] and the oriental hornet [27]. Queen-destined bumblebees receive large quantities of diet during the final instar, resulting in both increased size and an elongated growth period [28,29]. In these examples, diet quantity elevates juvenile hormone, and triggers the development of reproductive potential [30]. The fact that elevated juvenile hormone also induces queen development in honeybees [31,32] suggests a conserved mechanism in which reproductive status is regulated by diet and mediated by juvenile hormone. The importance of food quantity in other bee species suggests that diet quantity may contribute to caste determination in honeybees.

Here, we test the relative contributions of diet quantity and quality during caste determination in honeybees. Larvae were reared *in vitro* on diets varying in quality (protein and carbohydrate proportion) and quantity in a factorial design. Using this experimental design, we test the effect of quantity and quality simultaneously. Our results indicate that diet quantity influences queen differentiation in honeybees.

## 2. Material and methods

### (a) Artificial rearing

*Apis mellifera* larvae were collected from nine hives near Fargo, Cass County, North Dakota during a three-week period in the summer of 2015. Hives were supplemented with pollen patties (Mann Lake, MN, USA) and a 1 : 1 sucrose–water solution (Brushy Mountain Bee Farm, NC, USA) during poor foraging conditions. First instar larvae (0–21 h old) were transferred into 24-well cell culture plates (Falcon, Corning, Durham, NC) and placed onto 10 µl of diet. The 24-well plates were stored inside a modulator incubator chamber (Billups-Rothenberg, del Mar, CA, USA). Larvae were kept at a constant 34°C, darkness and relative humidity (RH) of 96% using potassium sulfate ($K_2SO_4$) [33]. Larvae were fed according to treatment in a factorial design of nine diet qualities and eight quantities with an additional ad libitum treatment for the medium diet, as described in the following sections. At the prepupal stage they were moved into 24-well cell culture plates containing Kim-wipes (Kimtech Science, USA) sterilized in EtOH [34]. Pupae were maintained at a constant 34°C, darkness and 75% RH using NaCl until adult eclosion. Adults were stored at −20°C.

### (b) Diet treatments

The study consisted of 72 treatment groups: nine diet qualities (table 1) combined with eight diet quantities in a factorial design. Additionally, an ad libitum quantity treatment was added using the medium-protein medium-carbohydrate diet (table 1). Each 24-well culture plate was randomly assigned to a diet quantity treatment. Within the plates, each row was assigned a diet quality treatment. Fresh diets were produced daily by homogenizing the ingredients for 10 min and warming in a 34°C water bath for 10 min before feeding. The volume of diet produced each day depended on the number of larvae in the study that were still in the feeding stage. For all the treatments, larvae were fed the same amount until the sixth day of development, while diet quality remained the same throughout development.

The reference diet (medium-protein medium-carbohydrate diet) was based on a previous study [35] which established the

diet induced worker development. The other eight diets were produced by altering carbohydrates (glucose and fructose) and protein content (royal jelly: Pure Royal Jelly eBeeHoney.com, Ashland, OH, USA) in a full factorial design. The same batch of royal jelly was used for all diets for the duration of the experiment. Glucose and fructose have been previously used to alter carbohydrate content in *in vitro* diets [33,35]. However, royal jelly is the only protein source for *in vitro* diets because adding non-royal jelly proteins such as casein significantly decreases survival [36]. Altering amounts of royal jelly to manipulate protein content also changes carbohydrate content because commercial royal jelly contains sugars in addition to proteins. Therefore, the protein, carbohydrate and water content of the royal jelly was quantified using a Bradford assay and differential scanning calorimetry (see the electronic supplementary material, methods). The royal jelly contained 12.35% protein, 27% carbohydrates and 56% water. These values were used to calculate the percentage of macronutrients in each diet.

## (c) Diet quantities

The lowest diet quantity (160 µl) was adopted from previous *in vitro* methods because this quantity produces workers [35]. Quantity was increased by 30 µl increments from 160 µl to 370 µl to produce the other treatments. There was an additional ad libitum treatment in which larvae were fed an excess of what they could consume. All larvae were fed the same amount during the first 5 days of development: day 1: 10 µl, day 2: 10 µl, day 3: 20 µl, day 4: 30 µl and day 5: 40 µl, totalling 110 µl of diet over the 5 days. During the sixth day of development, larvae were fed different amounts depending upon the diet quantity treatment so that total diet quantity ranged from 160 µl to 370 µl. In the ad libitum treatment, larvae were fed 200 µl per day until gut purge. Following gut purge, individuals were moved into pupation plates.

## (d) Morphometrics

Adult morphometrics can separate and classify castes, even when ovariole number and spermathecal size are excluded [37]. The mandibles, basitarsus and head were dissected from adults (see the electronic supplementary material, table S1 for sample sizes by treatment) and photographed. Morphometric measurements included total body wet weight, width and length of the basitarsus, width and length of the mandible, and width and length of the head (figure 1). IMAGEJ software was used for measurements. *In vitro* reared adults were compared to two reference populations: commercially reared queens (Wildflower Meadows, Southern CA, USA) and hive-reared workers collected from the research hives in early spring.

## (e) Data analysis and presentation of data

Statistical analyses were performed using R v.3.1.3 (R Core Team) [38]. Use of additional R packages are reported below where appropriate.

### (i) Principal component analysis

A principal component analysis (PCA) was used to categorize an individual as a queen, worker or intercaste by comparing morphometric measurements between *in vitro* reared individuals and reference workers and queens. The principal components were calculated from hive-reared workers and

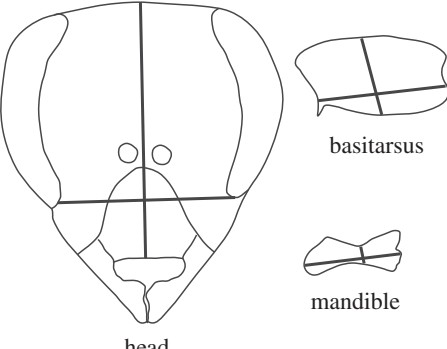

**Figure 1.** Measurements of honeybee head, basitarsus and mandible used in the morphometric analysis. Bold black lines represent the linear measurements of head length and width, basitarsus length and width, and mandible length and width.

commercially reared queens using the *prcomp* function in the *stats* package, and the *predict* function was used to produce principal components for the *in vitro* reared individuals. The assumptions for sphericity, sample adequacy and determinant of the matrix were tested and met. Principle component 1 (PC1) was used for downstream analysis.

### (ii) Clustering analysis

Clustering is a statistical analysis for group classification and was used to determine if *in vitro* reared individuals were grouped with reference queens or workers. Linkage distances were calculated using the complete method for hierarchical clustering [39]. The cluster analysis was performed using the *hclust* function in the *stats* package. The results were graphed using the *ColorDendrogram* function within the *sparcl* package [40]. The optimal number of clusters was calculated using the K-means clustering method.

### (iii) Measure of contribution of diet quantity and quality to principal component 1

The influence of diet quality and quantity on PC1 was compared for the 72 treatments, excluding the single ad libitum treatment. A generalized linear mixed model (GLMM) was performed using the *lme4* package [41]. PC1 was the dependent variable and diet quantity and diet quality (protein, carbohydrate and water proportion) were the independent variables (table 1). Hive was treated as a random effect because bees from different hives are expected to differ in size and shape because of parental genetics. The assumptions of collinearity, independence of data and normality were tested and met for the model.

## 3. Results

## (a) Principal component analysis

PCA was used to distinguish reference queens from reference workers based on six morphological measurements and body mass (figure 1). The PCA revealed a clear separation between reference workers and queens (figure 2), confirming morphometrics are a useful metric for caste classification. The results of the PCA also indicate that PC1 explains 64.42% of the variation between reference workers and queens, whereas principal component 2 (PC2) explained only 16.14% of

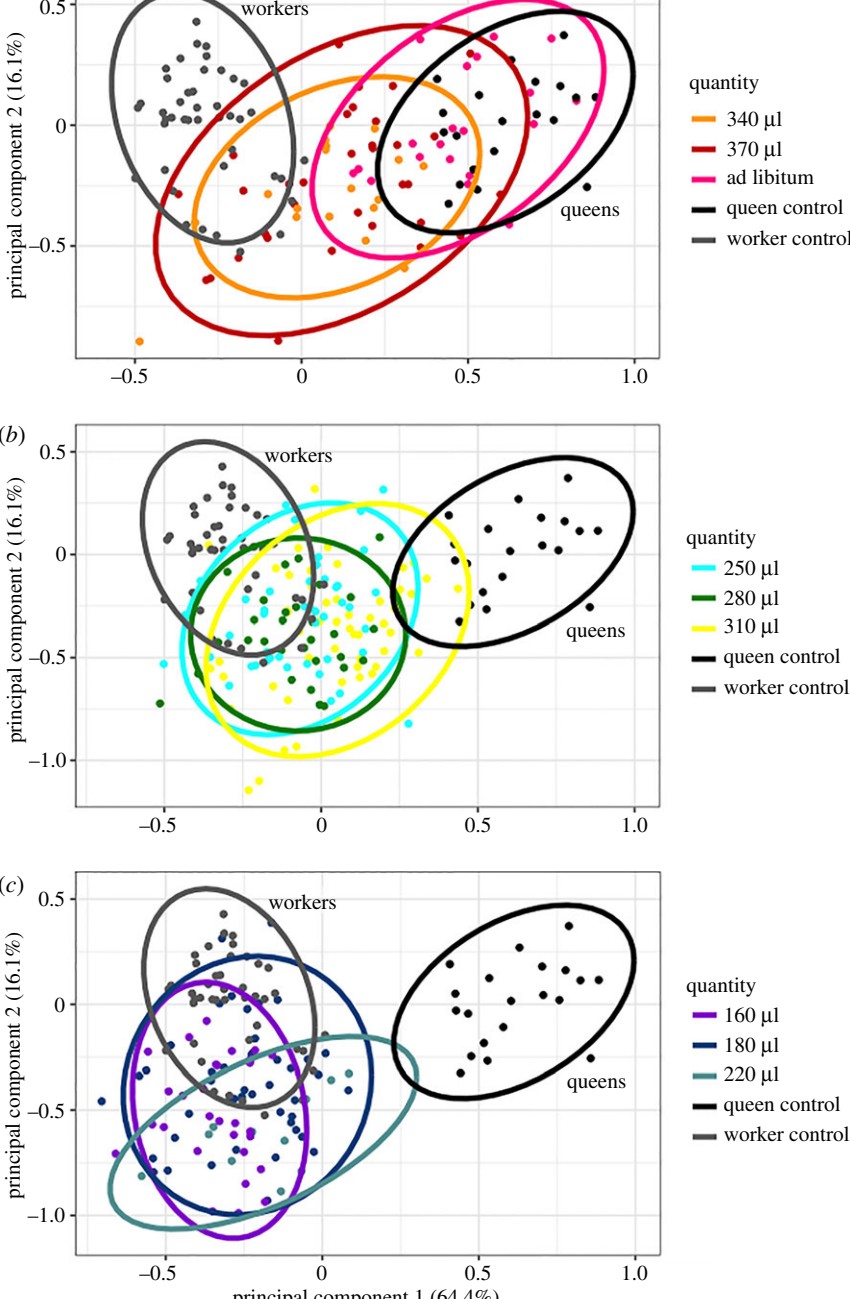

**Figure 2.** Principal component analysis (PCA) on seven traits visualized by diet quantity. Hive-reared queens (queen reference) and workers (worker reference) populations were analysed to establish principal component 1 (64.4%) and principal component 2 (16.14%). *In vitro* reared bees were compared to those reference populations. Colours represent amount of food during the final larval stage. (*a*) Bees fed larger quantities (340 µl, 370 µl and ad libitum) of food. (*b*) Bees fed medium quantities of food (250–310 µl). (*c*) Bees fed low quantities of food (160–220 µl). Ellipses represent the 95% confidence interval. Sample sizes were: 160 µl = 29, 190 µl = 45, 220 µl = 11, 250 µl = 44, 280 µl = 30, 310 µl = 54, 340 µl = 18, 370 µl = 33, ad libitum = 20, queen reference = 20, worker reference = 43. (Online version in colour.)

the variation. Over 95% of the variation is explained by the first five components (PC1 = 64.42, PC2 = 16.14, PC3 = 8.78, PC4 = 4.34, PC5 = 2.82). The eigenvalues for PC1 were strongly influenced by adult weight and basitarsus length (table 2; electronic supplementary material, figure S1). PC2 was strongly influenced by head length (table 2; electronic supplementary material, figure S1). Because PC1 explained the majority of the variation between workers and queens, we used PC1 as a quantification of the queenliness of *in vitro* reared individuals. A total of 282 adults emerged from the *in vitro* treatments. Some treatments had high mortality (electronic supplementary material, table S1), particularly the low protein diets. The *predict* function in

the *stats* package was used to calculate PC1 and PC2 for the *in vitro* reared individuals. PC1 increases as the quantity of diet increases (figure 2), but there was no discernable pattern with diet quality treatment (electronic supplementary material, figure S2). Statistical clustering and general linear modelling were used test the significance of the relationship between PC1 and diet quantity and quality.

## (b) Clustering analysis

Once PC1 was calculated for all individuals, a cluster analysis was used to determine significant groupings between individuals. Specifically, the goal was to determine which *in vitro*

**Table 2.** Eigenvalues of seven morphological traits of the first three principal components.

| trait | PC1 (64.42%) | PC2 (16.14%) | PC3 (8.78%) |
|---|---|---|---|
| adult weight | 55.2574 | 3.0590 | 7.4515 |
| basitarsus width | 0.1742 | 5.1875 | 0.0474 |
| basitarsus length | 32.5912 | 2.3740 | 0.3369 |
| head width | 4.0156 | 0.5515 | 90.7756 |
| head length | 5.2167 | 79.7015 | 1.3122 |
| mandible width | 2.5115 | 0.1337 | 0.0745 |
| mandible length | 0.2334 | 8.9928 | 0.0020 |

reared individuals were grouped with queens or workers, versus which were determined to be intercaste. The K-means clustering revealed that three clusters were optimal because three minimized the within cluster variation (40.3%), compared to four clusters (47.4%). The high quantity treatments occupied the cluster with commercially reared queens while the low quantity treatments occupied the worker cluster (table 3). The cluster analysis distinguished the reference workers and queens into separate groups, while the *in vitro* individuals occupied all three clusters. Of the *in vitro* reared bees, 81 clustered with commercial queens, 75 clustered with hive-reared workers and 128 formed an intercaste cluster (table 3). The high quantity treatments produced a high proportion of queens. For example, the ad libitum treatment produced 100% (20 out of 20) queens, and the 370 µl treatment produced 58% queens (19 out of 33) (table 3). In the lower quantity treatments (220, 190, 160 µl), only three queens were produced. Clustering with the *sparcl* package produced a dendrogram with reference queens clustering with individuals from high quantity treatments (figure 3). By contrast, the diet quality treatments of *in vitro* reared bees were more evenly distributed across clusters, with 19–39% clustering with queens for the high and medium protein diets (electronic supplementary material, table S2).

### (i) Effect of diet quantity and quality on final adult caste

A GLMM was used to determine whether PC1 was significantly influenced by diet quantity for *in vitro* reared individuals. The individuals from the ad libitum treatment were not included in the model because the treatment was not part of the factorial design. The GLMM results indicate quantity has a significant influence on PC1 ($p < 0.001$; table 4), whereas quality (protein, carbohydrate and water proportion) does not. Interaction terms were tested and excluded because they were not significant. The overall model explained 38.8% of the variation in PC1. Because the quality variables were not significant, the $R^2$ of 38.8% indicates that diet quantity, together with the random effect of hive origin, is explaining that amount of variation in PC1. In order to further examine the contributions of quantity and quality, a separate GLMM was run to test the influence of protein to carbohydrate ratio and quantity. Quantity was significant ($p < 0.0001$), but protein to carbohydrate ratio was not ($p = 0.244$). To visualize these relationships, diet quantity was plotted (figure 4), and increasing diet quantity had a significant effect on queenliness ($p < 0.0001$,

$R^2 = 0.3814$). The effects of different quality metrics (protein content, carbohydrate content, water content) were not significant (figure 5). However, protein to carbohydrate ratio did have a significant effect ($p = 0.0278$) when quantity was not taken into account. When each diet treatment was tested separately, quantity consistently had a positive influence on queenliness (electronic supplementary material, figure S3).

## 4. Discussion

Eusocial insects are an excellent example of phenotypic plasticity because they have reproductive and non-reproductive castes [1]. In many bee species, environmental cues drive female larvae into irreversible queen or worker developmental pathways [10]. These cues vary by species among eusocial Hymenoptera; however, only a few studies have fully characterized the cues that drive this divergence [25,26,28,29]. In honeybees, the cues driving caste bifurcation remain enigmatic [42]. Our goal was to determine the nutritional factors regulating caste, specifically testing whether diet quantity has a significant role in queen differentiation.

### (a) Increasing diet quantity induces queen traits

Diet quantity had a significant influence on adult caste. Larger quantities of diet increased the queenliness of adults regardless of diet quality. All of the ad libitum individuals (20 out of 20) clustered with the commercially reared queens, even though they were reared on a medium quality diet. Most of the 370 µl treatment (19 out of 33) clustered with commercially reared queens, while no queens were produced in the lowest quantity treatment (160 µl), independent of diet quality. In the GLMM, diet quantity significantly influenced PC1, our measurement of queenliness. Our study is, to our knowledge, the first to directly test the role of diet quantity on honeybee queen development, and observations of nurse behaviour support our findings. Nurse bees provide queen larvae with more food than workers throughout development [3,6]. When emergency queen rearing is necessary, nurses increase worker-destined larvae cell size in order to rear queens [2]. Increased cell size probably allows nurse bees to increase provisioning for queen rearing.

Traditionally, caste is empirically determined based upon reproductive metrics, such as ovariole number and spermatheca size [43]. While these metrics assess reproductively viability of queens, the measurements are time-consuming and must be done immediately after queens are collected. Our study resulted in 282 *in vitro* reared adults, and dissecting individuals immediately upon eclosion was not feasible. Other studies have used PCA on morphology to overcome the challenge of depending on internal traits. De Souza *et al.* [37] determined that morphometrics could successfully distinguish intercastes from queens and workers. Like De Souza *et al.* [37] our study did not include binary traits like the presence or absence of corbiculae and mandible notches because binary traits cannot be used in a PCA. In our study, PC1 explained 64.4% of the variation between reference queens and workers, which is more of the variation between workers and queens seen by De Souza *et al.* [37] (PC1 = 32.71%). Our PC1 was probably higher because we performed the PCA on just the reference queens and workers, and then used a prediction model to determine the principal components for the *in vitro*

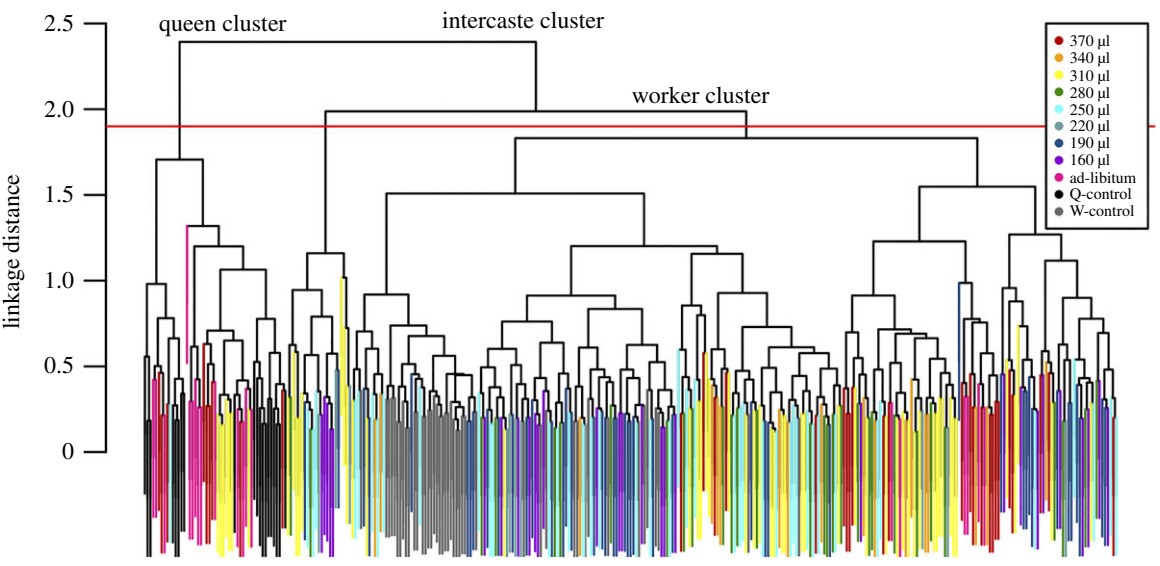

**Figure 3.** Dendrogram using ward linkage method for hierarchal clustering. Each colour represents either a quantitative treatment or a queen/worker control. The red line represents the significance threshold, which established three significant clusters. The three clusters are labelled worker, queen and intercaste based on the majority of the individuals occupying each cluster. (Online version in colour.)

**Table 3.** Number of individuals found in each hierarchical cluster determined by ward linkage.

| cluster | queen control | ad libitum | 370 µl | 340 µl | 310 µ | 280 µl | 250 µl | 220 µl | 190 µl | 160 µl | worker control |
|---|---|---|---|---|---|---|---|---|---|---|---|
| 1 | 20 | 20 | 19 | 8 | 20 | 3 | 8 | 1 | 2 | 0 | 0 |
| 2 | 0 | 0 | 12 | 7 | 26 | 18 | 17 | 9 | 20 | 19 | 0 |
| 3 | 0 | 0 | 2 | 3 | 8 | 9 | 19 | 1 | 23 | 10 | 43 |
| in queen cluster | 100% | | 58% | 44% | 37% | 10% | 18% | 9% | 4% | 0% | |

**Table 4.** Results of generalized linear mixed model. (The effects of diet quantity and quality (protein and carbohydrate proportion) were tested on principal component 1, which explains 64.42% of the variation between the reference workers and queens.)

| | principle component 1 (64.42%) | | |
|---|---|---|---|
| | $\beta$ | *t*-value | *p*-value |
| *fixed parts* | | | |
| (intercept) | −710.2 | 1.161 | 0.247 |
| total quantity | 0.002 | 12.047 | $p < 0.0001$ |
| protein proportion | −8.884 | 1.161 | 0.247 |
| carbohydrate proportion | −7.109 | 1.163 | 0.246 |
| water proportion | −7.309 | 1.162 | 0.246 |
| *random effects* | | | |
| hive | 0.009 | | |
| residual | 0.185 | | |
| $N_{Hive}$ | | 8 | |
| observations | | 264 | |
| $R^2$ | | 0.388 | |

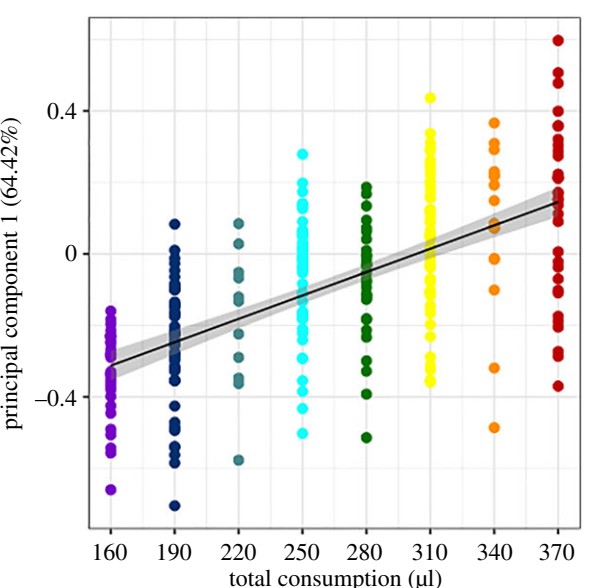

**Figure 4.** Influence of diet quantity on principal component 1 ($p < 0.0001$, $R^2 = 0.3814$). (Online version in colour.)

reared individuals. Morphometrics, including live weight, highly correlate with reproductive metrics, and are excellent predictors of reproductive viability of queens [44].

Many social hymenopterans use diet quantity to regulate caste determination [22,24–27,29]. Quantity controls caste in bees including bumblebees [28,29] and stingless bees [45]. Honeybees were one of the few exceptions with quality thought to play the primary role. Similar to honeybees, workers of other species increase cell size, and provision

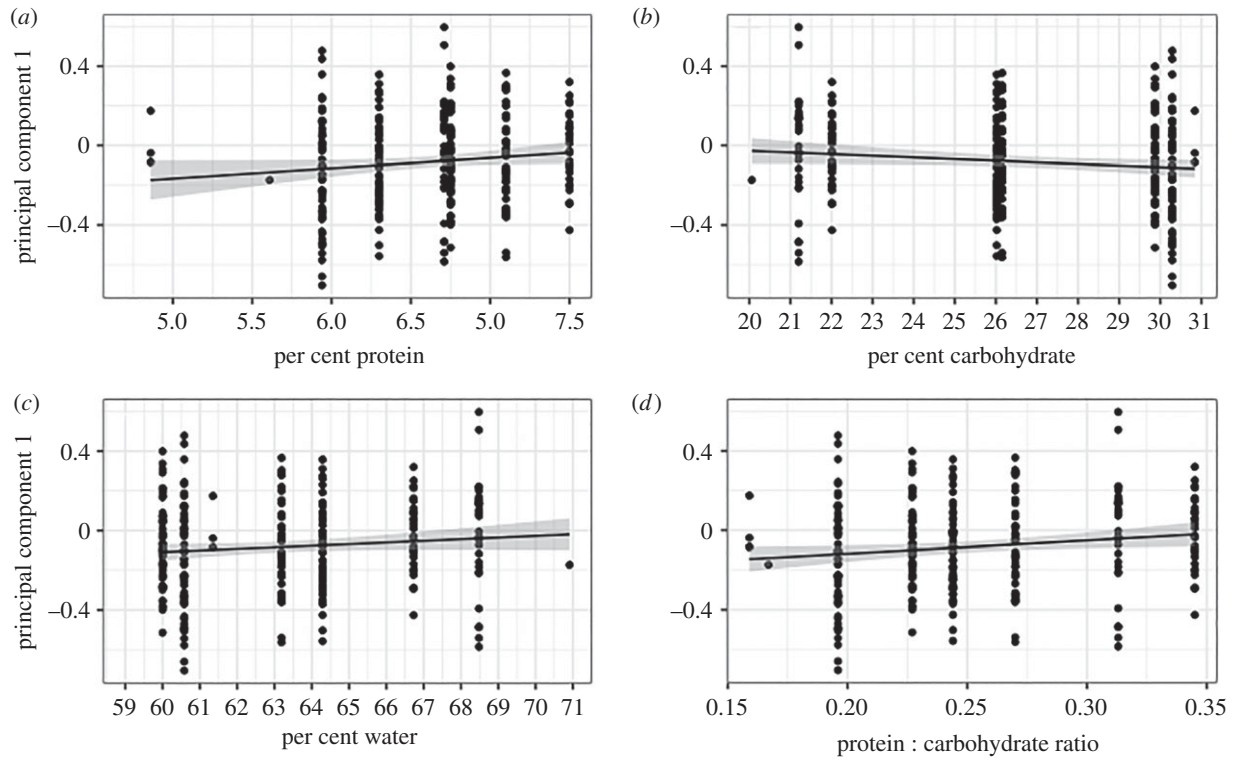

**Figure 5.** Influence of diet quality on principal component 1. (*a*) Protein proportion ($p = 0.0596$, $R^2 = 0.0097$), (*b*) carbohydrate proportion ($p = 0.0621$, $R^2 = 0.0095$), (*c*) water proportion ($p = 0.0997$, $R^2 = 0.0065$), and (*d*) protein to carbohydrate ratio ($p = 0.0278$, $R^2 = 0.0146$).

more diet in order to raise queens [46]. In *Bombus*, diet quantity elevates juvenile hormone in queen-destined larvae [29], indicating a conserved endocrine mechanism across bees [32]. In honeybees, the insulin pathways link nutritional status to juvenile hormone production [31,47–49], with the involvement of target of repamycin [31,49,50]. Ostensibly, these pathways can either trigger juvenile hormone release from the corpora allata [32] or be triggered by juvenile hormone, as observed during external application of the hormone [31]. Our findings that quantity regulates caste in honeybee suggests a conserved regulator of caste determination in bees.

## (b) Diet quality did not influence caste determination

We found that quantity, not protein proportion, carbohydrate proportion, nor water content, had a significant influence on final adult caste. This result is counter to previous studies, which identified these macronutrients as influencing caste [15,18,20]. Our results probably differ from previous studies because ours was, to our knowledge, the first study to systematically regulate consumption. Our results indicate these qualitative components do not seem to determine caste when food quantity is controlled. High carbohydrate diets produced more queens when diet quantity is not controlled [33]. When quantity was accounted for, Aupinel *et al.* [35] could not distinguish queen development between high and low carbohydrate diets, results that are corroborated by our study. Royalactin (MRJP-1) [15] and Major Royal Jelly Protein-3 (MRJP-3) [16] are considered the main royal jelly components influencing queen development, but neither study controlled for diet quantity. Our results indicate these qualitative components do not seem to determine caste when food quantity is controlled, and these qualitative factors should be reassessed to determine whether or not they play an important role in caste development.

Proteins have been widely studied for their role in caste determination, specifically royalactin (MRJP-1) [15] and MRJP-3 [16]. While studies initially showed these proteins were key regulators of caste [15,16], follow up studies have failed to reinforce these initial findings [51,52]. For example, Buttstedt *et al.* [51] added different amounts of monomeric royalactin *in vitro* and found royalactin did not induce queen development. Moreover, the major difference between the studies were Buttstedt *et al.* [51] controlled for diet quantity while Kamakura [15] did not. Since then, studies have revealed royalactin has multiple functions: the monomeric MRJP-1 is necessary for basic growth and development and the oligomeric MRJP-1 binds with 10-hydroxy-2-decenoic acid to increase royal jelly viscosity [53]. A current limitation of *in vitro* rearing is the inability to culture honeybee larvae on artificial diets components, such as casein [36], that would allow us to isolate the function of specific compounds. MRJPs have basic functions for larval growth and development, but it is unlikely one MRJP or a specific macronutrient determines caste alone.

Diet quality is important for juvenile growth, development and survival [34,35,54]. Growth is the result of not one but many interacting nutrients, including but not limited to proteins and carbohydrates. When protein and carbohydrate content are altered under controlled diet quantities, as done under a geometric framework for nutrition, growth rate, development and survival are affected, particularly on low protein diets [54]. We observed a similar result of high mortality on low protein diets. However, the proportion of proteins and carbohydrates do not appear to determine caste in honeybees. Carbohydrates can act as phagostimulants across insects [19,55–57], encouraging greater food consumption when present in high ratios. Sugars alone excite chemoreceptors as much as royal jelly [58]. Royal jelly contains significantly more carbohydrates than worker jelly [6,59]; thus,

carbohydrates may be the phagostimulant that increases food consumption in queen-destined larvae. Royal jelly proteins are thought to regulate caste, but royal jelly and worker jelly do not always differ in the proportion of MRJPs. One study found royal jelly had higher protein content than worker jelly [59], whereas others reported lower protein content [8,60]. Protein and carbohydrates have important roles for larval growth and development, but the proportion of their components may not be the important difference between the larval diets of queens and workers.

In this study, we chose to focus on proteins, carbohydrates and water because previous studies showed these diet components had the strongest influence on queen development [15,18,20], even though royal jelly contains lipids and micronutrients [6] which also differed between our diets. Despite these limitations, our study corroborates other studies that have indicated relative protein content does not control queen differentiation [51,52]. Our results indicate diet quantity has a larger role on queen development than quality, and future studies should control for diet quantity when systematically testing macronutrients on caste development.

## 5. Conclusion

Nutritional status regulates many insect polyphenisms, such as the horns of dung beetles, and increased production of non-worker castes in termites [61–63]. A biological active substance in royal jelly has been thought to determine caste in honeybees since the late 1890s [3,6,21]. Many studies have evaluated caste determination both *in vivo* and *in vitro* [31,47,49,50], and while these studies offer important insights, the influence of diet quantity has not been explicitly tested. Our results indicate diet quantity is a significant factor in caste determination. Nutritional stress is a simple and elegant mechanism for controlling reproductive potential. In social vertebrates, nutritional stress regulates reproductive status [64]. During the evolution of eusociality in wasps, lower nutritional status in workers probably regulated their non-reproductive status [65]. In solitary bee species, which represent the ancestral state with respect to sociality [66], all female larvae become reproductive adults. Therefore, caste determination is a loss of reproductive potential in worker larvae, not necessarily a gain of reproductive potential caused by ingestion of a special type of food. During the evolution of eusociality, the loss of reproductive potential in workers coincides with the development of multiple morphological and behavioural traits that support hive fitness. miRNAs and p-coumaric acid found in pollen inhibit ovary development and are only present in the diet of worker-destined larva [67,68]. In our study, low food quantities inhibited queen development. The third day has long been thought the critical window for queen determination [12], but our results suggest reproductive status can be retained up to the sixth day if sufficient food is present. Thus, our study and [68] suggests honeybee larvae become workers through dietary suppression, such as food restriction and p-coumaric acid. In this framework, the mechanism that underlies eusociality in bees is food restriction, which suppresses reproduction in the worker caste.

**Data accessibility.** Data available from the Dryad Digital Repository: https://doi.org/10.5061/dryad.h44j0zpgc [69].

**Author contributions.** G.P.S. and J.H.B. designed the experiment. G.P.S. performed the experiment and statistical analyses. J.H.B. and G.D.Y. contributed statistical expertise. G.P.S., J.H.B. and G.D.Y. wrote the manuscript.

**Competing interests.** We declare we have no competing interests.

**Funding.** This work was supported by a grant from the National Science Foundation (grant nos NSF IOS-155794 and NSF-RII-1826834), resources from the USDA ARS Insect Genetics and Biochemistry (grant no. 3060-21000-041-00D) and funding from the Department of Biological Sciences at North Dakota State University.

**Acknowledgements.** We would like to thank M. Palmersheim and E. Jensen for contributing to data collection. A. Rajamohan helped develop experimental procedures and contributed to data collection. J. Rinehart and M. Larson provided technical support. B. Heidinger provided comments on the manuscript. We thank four anonymous reviewers for their comments.

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
