## [Reviewer comments · Proceedings of the Royal Society B: Biological Sciences]

Review History

RSPB-2019-1944.R0 (Original submission)

Review form: Reviewer 1

Recommendation

Accept with minor revision (please list in comments)

Scientific importance: Is the manuscript an original and important contribution to its field?

Excellent

General interest: Is the paper of sufficient general interest?

Good

Quality of the paper: Is the overall quality of the paper suitable?

Good

Is the length of the paper justified?

Yes

Should the paper be seen by a specialist statistical reviewer?

No

Do you have any concerns about statistical analyses in this paper? If so, please specify them explicitly in your report.

No

It is a condition of publication that authors make their supporting data, code and materials available - either as supplementary material or hosted in an external repository. Please rate, if applicable, the supporting data on the following criteria.

Is it accessible?

N/A

Is it clear?

N/A

Is it adequate?

N/A

Do you have any ethical concerns with this paper?

No

Comments to the Author

See attached file. (See Appendix A)

Review form: Reviewer 2

Recommendation

Major revision is needed (please make suggestions in comments)

Scientific importance: Is the manuscript an original and important contribution to its field?

Acceptable

General interest: Is the paper of sufficient general interest?

Acceptable

Quality of the paper: Is the overall quality of the paper suitable?

Acceptable

Is the length of the paper justified?

Yes

Should the paper be seen by a specialist statistical reviewer?

No

Do you have any concerns about statistical analyses in this paper? If so, please specify them explicitly in your report.

No

It is a condition of publication that authors make their supporting data, code and materials available - either as supplementary material or hosted in an external repository. Please rate, if applicable, the supporting data on the following criteria.

Is it accessible?

No

Is it clear?

Yes

Is it adequate?

No

Do you have any ethical concerns with this paper?

No

Comments to the Author

Slater et al. challenge conventional wisdom about how caste is determined in honey bees. They do this through in vitro rearing of honey bees under different diet regimes varying in quality and quantity in a fully factorial design, characterizing the resulting adults as queens, workers, or intercastes based on external morphology. Caste determination is an important feature of eusocial insects and honey bees have played a major role in understanding social behavior, and thus the study has relevance to the field. However, I have a number of concerns about the methods and interpretation of the results that prevent me from accepting their conclusions at face value.

The diet treatments used are based upon diets previously shown to induce worker development in honey bees (the lowest diet quantity and the medium-protein medium-carbohydrate diets are cited from another study as worker-inducing). The other diet treatments involved larger quantities relative to this worker reference diet, as well as mixes of royal jelly with glucose and fructose to vary "quality". I would like to have seen a diet treatment of entirely royal jelly, or at least diets that were closer in "quality" to pure royal jelly. That aside, the diets they did use led to some surprising effects. First, the lowest quantity diet produced more intercastes than workers. Table 3 doesn't break down the numbers of individuals in each cluster by quality (these data need to be available in a supplemental table!), but seeing 19 intercastes and 10 workers eclose from larvae reared on the lowest quantity diet is surprising, and to me indicates that quality MAY matter, as the variation in caste must be coming from somewhere. I see no discussion about why, even at the lowest quantity tested, there is so much variation in phenotype. Is in vitro rearing wildly different between labs such that the same diet in one lab leads to very different results in another? I worry about the relevance of the findings with respect to natural caste determination in honey bees (as opposed to lab reared bees).

I also struggle to separate quality from quantity in their results and interpretation. The authors suggest that there is no "biological active substance" in royal jelly that makes a queen. I don't think this conclusion can be made with the present data. There MAY be some component of royal jelly that is necessary, above some threshold, in order to develop queen traits. This threshold could be reached by a lower quantity of a high "quality" diet (pure royal jelly would have more than diluted royal jelly) OR by a larger quantity of a low "quality" diet. In other words, "quantity has a quality all its own". There may be a compensatory effect of quantity that makes up for a lack in quality, but that doesn't mean that diet quality doesn't have an effect with natural diets under natural conditions. Further, the range in diet quantity tested appears, at least superficially, a lot greater than the range in quality of diets (e.g., 2.3-fold variation in quantity, but only 1.5-fold variation in protein content). This, combined with the fact that a queen-like diet wasn't tested (pure royal jelly), makes me wonder if an effect of quality would have been seen with a more realistic (fully worker to fully queen) range of diet qualities.

I understand that there are limitations in what can be tested, and the authors mention that casein and other more artificial diets cannot be used for larval rearing of honey bees. However, using royal jelly as a protein source comes with a range of complications, some of which could at least be taken into consideration in the discussion of this manuscript. Other authors have used heat to denature the proteins in royal jelly and have seen effects on caste determination (inability to rear queens). Can the authors speak to these previous findings? Did the authors experiment with trying to "deactivate" any putative "biological active" substances in the royal jelly? I struggle to

view royal jelly as just a source of protein, and reducing its content to a percent protein, carbohydrate, and water seems unrealistic.

In addition, I have a number of minor questions, comments, and suggestions for improvements:

Line 34: "caste" should be "castes"

Line 40 and 234: not all eusocial insects show environmentally-determined castes (for example, some stingless bees have genetic caste determination)

Line 55: "gland" should be "glands", or maybe "glandular secretions"?

Line 59: no comma before "all"

Line 62: how much diet do workers get? 1.5 g for queens, would be nice to compare with workers, and also compare the diets used in this study (given in volumes) to these numbers

Line 69: add "an" before "elongated"

Line 71: add "also" before "induces"

Lines 105-107: Discussion of a feeding stage makes me wonder- can larvae prolong their feeding stage? Could larvae receiving lower quantity/quality diets extend the time between instars or prior to pupation? In this way, could they make up for any deficits due to lower quantity/quality?

Line 114: "can be mixed into..." better written as "have been previously used to alter carbohydrate content of in vitro diets"

Lines 120-121: Was the same stock of royal jelly used for all treatments? Are the protein, carbohydrate, and water contents of this royal jelly similar to others used? How consistent are the macronutrients of royal jelly?

Morphometrics section: For confirmatory purposes, it would have been nice if variation in ovariole number and spermathecal size were examined in a subset of individuals.

Line 160: "hierarchal" should be "hierarchical"

Line 179: remove "The"

Line 180: seven morphological measurements mentioned, but only see 6 in figure (eventually realized the 7th was adult weight, but not clear until later)

Fig. 2- sample sizes should be indicated in legend; difficult to discern all groups with colors used, maybe rather than a ROYGBIV color scheme a gradient of one color for quantity would be more clear? In addition, it would be nice to see a similar figure which colors individuals by quality rather than quantity which would presumably show a lack of grouping by quality?

Fig. 2 and 3- legends are not consistent in their ordering, which threw me initially

Line 192: space in "PCland"

Fig. 3- very difficult to parse the individuals and colors; can leaves of the same color be combined into larger color blocks? Again, colors are difficult to discern with respect to quantity groupings

Table 3- at LEAST in a supplemental table, I really desperately want to see a breakdown of queen/worker/intercaste clusters by quality. Also, why not label clusters as

queen/worker/intercaste as opposed to numbers?

Lines 207-209: While the ad libitum treatment produced all queens, I find 58% from the highest quantity diet lower than expected if quantity is so important. One big difference, if I am interpreting the methods correctly, is that the ad libitum group received more diet throughout larval development, while the other diet treatments only varied in the final stages of development. Wouldn't this indicate that timing is also important, in addition to quantity? Could the authors speak to this?

Line 220: is "quality" meant to be "quantity" here? Either an important typo or I'm confused

Line 226: I don't like a sentence that starts with "And,"

Line 233-234: Division of labor is not really why I view eusocial insects as a good example of phenotypic plasticity; strange phrasing at the very least

Lines 236-237: which studies have "fully characterized" the cues that drive caste divergence? Citations?

Line 245: 19/33 is technically most, sure, but there are plenty of the high quantity individuals that did NOT cluster with commercially-reared queens, perhaps indicating that the range of quantities/qualities used is not representative of the full range of relevant diets to compare queen and worker caste determination

Lines 256-257: do ovariole number and spermathecal size measurements really have to be done immediately after collection? Why couldn't individuals be frozen first? And as far as mortality, I believe the authors killed them anyway, so dissection should have been possible

Line 271: "Many social hymenopterans use diet quantity to regulate caste determination". I would argue that obviously honey bees do this as well- queen honey bees are fed much more...

Line 277: "links" should be "link"

Line 278: missing word "of" before TOR

Line 307: MRJP not MJRP

Line 309: "may not differ" is strange phrasing; "do not always differ"?

Line 310: are the authors referring to all proteins, or specifically MRJP1 and 3?

Line 313: extra "of"

Line 328: respect "to" sociality, not "of"

Line 346: "Ours" should be "Our"

Line 346: I disagree that diet quantity is "unacknowledged" with respect to honey bee caste determination- anyone who has ever seen a queen cell will attest to the fact that certainly, queens receive a lot more food than workers

Decision letter (RSPB-2019-1944.R0)

27-Sep-2019

Dear Dr Bowsher:

I am writing to inform you that your manuscript RSPB-2019-1944 entitled "Diet quantity influences caste determination in honey bees (*Apis mellifera*)" has, in its current form, been rejected for publication in Proceedings B.

This action has been taken on the advice of referees, who have recommended that substantial revisions are necessary. With this in mind we would be happy to consider a resubmission, provided the comments of the referees are fully addressed. However please note that this is not a provisional acceptance.

Sincerely,
Dr Sasha Dall
<mailto:proceedingsb@royalsociety.org>

Associate Editor
Board Member: 1
Comments to Author:

I have obtained two thoughtful and thorough reviews of the manuscript. Both reviewers see significant value in this manuscript (though one reviewer is considerably more enthusiastic than the other) and I fully agree in that assessment. I therefore encourage the authors to revise the manuscript in the light of both reviews. Further, I would like to add two comments: (i) Figure 1 could be improved in all aspects, from light and contrast to focus, perhaps by generating composite images from z-stacks; (ii) the entire introduction and discussion are eusocial hymenoptera focused. But there are other eusocial insects out there for which dietary information is available (e.g. termites), and there are non-eusocial insects out there which are nonetheless polyphenic and both diet quality and quantity effects have been assessed (e.g. dung beetles). Broadening the discussion accordingly may allow the authors to place their results into a broader context to further explore the distinctness (or lack thereof) of caste determination in honey bees.

Reviewer(s)' Comments to Author:

Referee: 1

Comments to the Author(s)

See attached file

Referee: 2

Comments to the Author(s)

Slater et al. challenge conventional wisdom about how caste is determined in honey bees. They do this through in vitro rearing of honey bees under different diet regimes varying in quality and quantity in a fully factorial design, characterizing the resulting adults as queens, workers, or intercastes based on external morphology. Caste determination is an important feature of eusocial insects and honey bees have played a major role in understanding social behavior, and thus the study has relevance to the field. However, I have a number of concerns about the methods and interpretation of the results that prevent me from accepting their conclusions at face value.

The diet treatments used are based upon diets previously shown to induce worker development in honey bees (the lowest diet quantity and the medium-protein medium-carbohydrate diets are cited from another study as worker-inducing). The other diet treatments involved larger quantities relative to this worker reference diet, as well as mixes of royal jelly with glucose and fructose to vary "quality". I would like to have seen a diet treatment of entirely royal jelly, or at least diets that were closer in "quality" to pure royal jelly. That aside, the diets they did use led to some surprising effects. First, the lowest quantity diet produced more intercastes than workers. Table 3 doesn't break down the numbers of individuals in each cluster by quality (these data need to be available in a supplemental table!), but seeing 19 intercastes and 10 workers eclose from larvae reared on the lowest quantity diet is surprising, and to me indicates that quality MAY matter, as the variation in caste must be coming from somewhere. I see no discussion about why, even at the lowest quantity tested, there is so much variation in phenotype. Is in vitro rearing wildly different between labs such that the same diet in one lab leads to very different results in another? I worry about the relevance of the findings with respect to natural caste determination in honey bees (as opposed to lab reared bees).

I also struggle to separate quality from quantity in their results and interpretation. The authors suggest that there is no "biological active substance" in royal jelly that makes a queen. I don't think this conclusion can be made with the present data. There MAY be some component of royal jelly that is necessary, above some threshold, in order to develop queen traits. This threshold could be reached by a lower quantity of a high "quality" diet (pure royal jelly would have more than diluted royal jelly) OR by a larger quantity of a low "quality" diet. In other words, "quantity has a quality all its own". There may be a compensatory effect of quantity that makes up for a lack in quality, but that doesn't mean that diet quality doesn't have an effect with natural diets under natural conditions. Further, the range in diet quantity tested appears, at least superficially, a lot greater than the range in quality of diets (e.g., 2.3-fold variation in quantity, but only 1.5-fold variation in protein content). This, combined with the fact that a queen-like diet wasn't tested (pure royal jelly), makes me wonder if an effect of quality would have been seen with a more realistic (fully worker to fully queen) range of diet qualities.

I understand that there are limitations in what can be tested, and the authors mention that casein and other more artificial diets cannot be used for larval rearing of honey bees. However, using royal jelly as a protein source comes with a range of complications, some of which could at least be taken into consideration in the discussion of this manuscript. Other authors have used heat to denature the proteins in royal jelly and have seen effects on caste determination (inability to rear

queens). Can the authors speak to these previous findings? Did the authors experiment with trying to “deactivate” any putative “biological active” substances in the royal jelly? I struggle to view royal jelly as just a source of protein, and reducing its content to a percent protein, carbohydrate, and water seems unrealistic.

In addition, I have a number of minor questions, comments, and suggestions for improvements:

Line 34: “caste” should be “castes”

Line 40 and 234: not all eusocial insects show environmentally-determined castes (for example, some stingless bees have genetic caste determination)

Line 55: “gland” should be “glands”, or maybe “glandular secretions”?

Line 59: no comma before “all”

Line 62: how much diet do workers get? 1.5 g for queens, would be nice to compare with workers, and also compare the diets used in this study (given in volumes) to these numbers

Line 69: add “an” before “elongated”

Line 71: add “also” before “induces”

Lines 105-107: Discussion of a feeding stage makes me wonder- can larvae prolong their feeding stage? Could larvae receiving lower quantity/quality diets extend the time between instars or prior to pupation? In this way, could they make up for any deficits due to lower quantity/quality?

Line 114: “can be mixed into...” better written as “have been previously used to alter carbohydrate content of in vitro diets”

Lines 120-121: Was the same stock of royal jelly used for all treatments? Are the protein, carbohydrate, and water contents of this royal jelly similar to others used? How consistent are the macronutrients of royal jelly?

Morphometrics section: For confirmatory purposes, it would have been nice if variation in ovariole number and spermathecal size were examined in a subset of individuals.

Line 160: “hierarchal” should be “hierarchical”

Line 179: remove “The”

Line 180: seven morphological measurements mentioned, but only see 6 in figure (eventually realized the 7th was adult weight, but not clear until later)

Fig. 2- sample sizes should be indicated in legend; difficult to discern all groups with colors used, maybe rather than a ROYGBIV color scheme a gradient of one color for quantity would be more clear? In addition, it would be nice to see a similar figure which colors individuals by quality rather than quantity which would presumably show a lack of grouping by quality?

Fig. 2 and 3- legends are not consistent in their ordering, which threw me initially

Line 192: space in “PC1and”

Fig. 3- very difficult to parse the individuals and colors; can leaves of the same color be combined into larger color blocks? Again, colors are difficult to discern with respect to quantity groupings

Table 3- at LEAST in a supplemental table, I really desperately want to see a breakdown of queen/worker/intercaste clusters by quality. Also, why not label clusters as queen/worker/intercaste as opposed to numbers?

Lines 207-209: While the ad libitum treatment produced all queens, I find 58% from the highest quantity diet lower than expected if quantity is so important. One big difference, if I am interpreting the methods correctly, is that the ad libitum group received more diet throughout larval development, while the other diet treatments only varied in the final stages of development. Wouldn't this indicate that timing is also important, in addition to quantity? Could the authors speak to this?

Line 220: is "quality" meant to be "quantity" here? Either an important typo or I'm confused

Line 226: I don't like a sentence that starts with "And,"

Line 233-234: Division of labor is not really why I view eusocial insects as a good example of phenotypic plasticity; strange phrasing at the very least

Lines 236-237: which studies have "fully characterized" the cues that drive caste divergence? Citations?

Line 245: 19/33 is technically most, sure, but there are plenty of the high quantity individuals that did NOT cluster with commercially-reared queens, perhaps indicating that the range of quantities/qualities used is not representative of the full range of relevant diets to compare queen and worker caste determination

Lines 256-257: do ovariole number and spermathecal size measurements really have to be done immediately after collection? Why couldn't individuals be frozen first? And as far as mortality, I believe the authors killed them anyway, so dissection should have been possible

Line 271: "Many social hymenopterans use diet quantity to regulate caste determination". I would argue that obviously honey bees do this as well- queen honey bees are fed much more...

Line 277: "links" should be "link"

Line 278: missing word "of" before TOR

Line 307: MRJP not MJRP

Line 309: "may not differ" is strange phrasing; "do not always differ"?

Line 310: are the authors referring to all proteins, or specifically MRJP1 and 3?

Line 313: extra "of"

Line 328: respect "to" sociality, not "of"

Line 346: "Ours" should be "Our"

Line 346: I disagree that diet quantity is "unacknowledged" with respect to honey bee caste determination- anyone who has ever seen a queen cell will attest to the fact that certainly, queens receive a lot more food than workers

Author's Response to Decision Letter for (RSPB-2019-1944.R0)

See Appendix B.

RSPB-2020-0614.R0

Review form: Reviewer 2

Recommendation

Accept with minor revision (please list in comments)

Scientific importance: Is the manuscript an original and important contribution to its field?

Good

General interest: Is the paper of sufficient general interest?

Good

Quality of the paper: Is the overall quality of the paper suitable?

Good

Is the length of the paper justified?

Yes

Should the paper be seen by a specialist statistical reviewer?

No

Do you have any concerns about statistical analyses in this paper? If so, please specify them explicitly in your report.

No

It is a condition of publication that authors make their supporting data, code and materials available - either as supplementary material or hosted in an external repository. Please rate, if applicable, the supporting data on the following criteria.

Is it accessible?

Yes

Is it clear?

Yes

Is it adequate?

Yes

Do you have any ethical concerns with this paper?

No

Comments to the Author

I appreciate the authors' consideration of my previous feedback and feel the manuscript has been improved to a state worthy of publication. The finding that diet quantity, rather than quality, influences caste determination in honey bees is an important one that will influence the field of sociobiology.

I have just a few additional minor comments/changes to suggest:

Lines 28-30: "We found that larvae fed the largest quantities of diet were indistinguishable from hive reared queens, independent of the proportion of protein and carbohydrate in the diet"

- Not all larvae fed the largest quantities were indistinguishable from queens, right? Just a subset?
Unless "largest quantities" refers to ad lib, which I think is misleading because the abstract refers to 8 different quantities, and there are 8 plus ad lib.

Line 42: "queen-worker phenotypes"

- Queen and worker phenotypes?

Line 63: Is 1.5g the amount extra, or the total amount fed to the queen, not all of which is consumed? Not clear.

Line 73: should not be a comma after (31, 32) references

Line 94: "eight diet qualities and nine quantities" vs lines 25-26 "Larvae were reared in vitro on nine diets varying in amount of royal jelly and sugars, which were fed to larvae in eight different quantities."

Methods: I would add a sentence clarifying that the same royal jelly stock was used for all diets for the duration of the experiment

Line 132: "pupations" should be "pupation"

Line 150: "intercastes" should be "intercaste"

Line 211: "with 22-39% clustering with queens for the high and medium protein diets (Table S2)."

- For high and medium protein diets it appears the range is 19-39%, not 22-39%

Lines 217-218: "ad libitum treatment were not included in the model because the treatment was not factorial for QUANTITY"

- I assume the authors mean quality, not quantity here?

Line 248: "influences" should be "influenced"

Line 249: "first to directly test the role of diet quantity [on honey bee queen development]"

- I recommend adding the qualifier for clarity

Line 255: maybe more clear as "caste is empirically determined based upon"

Line 296: if MRJP-1 and MRJP-3 influence queen development in the absence of controlling quantity, does this mean these proteins may influence larvae to consume more food, thus leading to queenliness? I wonder if the authors could speculate about the role these proteins play in consumption.

Line 299: might as well use the MRJP-1 and MRJP-3 abbreviations (but throughout this paragraph the abbreviation is incorrectly MJRP rather than MRJP)

Line 302: Royalactin is capitalized here but not elsewhere, and the abbreviation has already been given twice so no need to do that again

Line 335: "tested" should be "testing"

There are typos and inconsistent formatting in the references; please check these carefully.

Decision letter (RSPB-2020-0614.R0)

17-Apr-2020

Dear Dr Bowsher

I am pleased to inform you that your manuscript RSPB-2020-0614 entitled "Diet quantity influences caste determination in honey bees (*Apis mellifera*)" has been accepted for publication in Proceedings B.

The referee(s) have recommended publication, but also suggest some minor revisions to your manuscript. Therefore, I invite you to respond to the referee(s)' comments and revise your manuscript. Because the schedule for publication is very tight, it is a condition of publication that you submit the revised version of your manuscript within 7 days. If you do not think you will be able to meet this date please let us know.

[http://datadryad.org/submit?journalID=RSPB&manu=\(Document not available\)](http://datadryad.org/submit?journalID=RSPB&manu=(Document not available)) which will take you to your unique entry in the Dryad repository. If you have already submitted your data to dryad you can make any necessary revisions to your dataset by following the above link.

Please see <https://royalsociety.org/journals/ethics-policies/data-sharing-mining/> for more details.

Sincerely,

Dr Sasha Dall

Associate Editor

Board Member

Comments to Author:

I have now received one review of your revised manuscript. I thank you for taking the reviewers' suggestions to heart during the revision process and agree with the remaining reviewer in recommending the manuscript for publication! The reviewer did list a few additional suggestions in the last reviewing round, however, and while they are all essentially minor there are quite a few of them and I trust that you will do your best to incorporate them in your final revisions. Thank you for submitting your work to the Proceedings and congratulations to fine contribution to the field.

Reviewer(s)' Comments to Author:

Referee: 2

Comments to the Author(s).

I appreciate the authors' consideration of my previous feedback and feel the manuscript has been improved to a state worthy of publication. The finding that diet quantity, rather than quality,

influences caste determination in honey bees is an important one that will influence the field of sociobiology.

I have just a few additional minor comments/changes to suggest:

Lines 28-30: "We found that larvae fed the largest quantities of diet were indistinguishable from hive reared queens, independent of the proportion of protein and carbohydrate in the diet"

- Not all larvae fed the largest quantities were indistinguishable from queens, right? Just a subset? Unless "largest quantities" refers to ad lib, which I think is misleading because the abstract refers to 8 different quantities, and there are 8 plus ad lib.

Line 42: "queen-worker phenotypes"

- Queen and worker phenotypes?

Line 63: Is 1.5g the amount extra, or the total amount fed to the queen, not all of which is consumed? Not clear.

Line 73: should not be a comma after (31, 32) references

Line 94: "eight diet qualities and nine quantities" vs lines 25-26 "Larvae were reared in vitro on nine diets varying in amount of royal jelly and sugars, which were fed to larvae in eight different quantities."

Methods: I would add a sentence clarifying that the same royal jelly stock was used for all diets for the duration of the experiment

Line 132: "pupations" should be "pupation"

Line 150: "intercastes" should be "intercaste"

Line 211: "with 22-39% clustering with queens for the high and medium protein diets (Table S2)."

- For high and medium protein diets it appears the range is 19-39%, not 22-39%

Lines 217-218: "ad libitum treatment were not included in the model because the treatment was not factorial for QUANTITY"

- I assume the authors mean quality, not quantity here?

Line 248: "influences" should be "influenced"

Line 249: "first to directly test the role of diet quantity [on honey bee queen development]"

- I recommend adding the qualifier for clarity

Line 255: maybe more clear as "caste is empirically determined based upon"

Line 296: if MRJP-1 and MRJP-3 influence queen development in the absence of controlling quantity, does this mean these proteins may influence larvae to consume more food, thus leading to queenliness? I wonder if the authors could speculate about the role these proteins play in consumption.

Line 299: might as well use the MRJP-1 and MRJP-3 abbreviations (but throughout this paragraph the abbreviation is incorrectly MJRP rather than MRJP)

Line 302: Royalactin is capitalized here but not elsewhere, and the abbreviation has already been given twice so no need to do that again

Line 335: "tested" should be "testing"

There are typos and inconsistent formatting in the references; please check these carefully.

Author's Response to Decision Letter for (RSPB-2020-0614.R0)

See Appendix C.

Decision letter (RSPB-2020-0614.R1)

27-Apr-2020

Dear Dr Bowsher

I am pleased to inform you that your manuscript entitled "Diet quantity influences caste determination in honey bees (*Apis mellifera*)" has been accepted for publication in Proceedings B.

Open Access

Paper charges

Sincerely,
Proceedings B
mailto: proceedingsb@royalsociety.org

Appendix A

This manuscript reports much needed work that examined the effect of food quantity on caste determination in honey bees. That royal jelly is the magic food enabling queen development has captured the imagination for centuries and is similarly dated. The belief that it contains one single, magic ingredient that effects the development switch is unwarranted.

Comments

Line 71. THE FACT THAT Elevated JH ALSO induces queen develop in honey bees, WHICH HAVE MORE COMPLEX SOCIALITY, suggests a conserved mechanism IN WHICH DIET WORKS THROUGH JH TO PRODUCE INDIVIDUALS WITH HIGHER REPRODUCTIVE POTENTIAL.

Or some other way of expressing explicitly what conserved you are thinking of.

Royal jelly is used in all treatments – so the all the other components in it are still there though in less amounts. Is this a problem?

For example, might there be sufficient royalactin to maintain pluripotency, in most treatments? See Nature Communications 2018 Wan et al. Honey bee royalactin unlocks conserved pluripotency in mammals.

243. It is amazing that ad libitum individuals do as well as commercially reared queens. This is such an important finding. Love it.

251. In addition to cell size, they change the orientation. This is apparently related to higher viscosity of RJ structure and viscosity (something in Insectes Soc. Recently). This may be another way quantity is controlled in the hive setting. Another mystery here.

278. Clarify that TOR is in the IIS pathway and EGFR is not? TOR is not independent of JH. Mutti (31) title says IRS and TOR pathways act via JH. In addition, add that JH was artificially applied in Mutti's case.

329. Yes a loss of reproductive function but so much more is involved, such as numerous other features associated brood care and foraging.

336. I disagree with that the study suggest that larvae are predetermined queens but for starvation and other abuse. Any (most) organisms have within them the information to develop robustly under that best of conditions – but those conditions rarely exist so it doesn't mean the most robust form is the default. Queen determination is a bifurcating developmental system in which nutritional cues are manipulated and assessed through multiple pathways to respond to conditions they experience. Both castes are highly specialized outcomes.

OTHER

The Bowsher lab has also published a paper on honey bee development and the geometric model of nutritional needs. I would love to see these findings integrated here. Should we expect this model to work when nutrition and its developmental outcomes are not unitary?

Misc .

Line 45, feed=fed

Line 220, diet quality. If you mean quality, than I am confused by this sentence.

And finally, are the percentages really significant to 2 decimal places?

Line 248. and instead of but

302. A result

313. Bee larvae (of) on – delete of.

Appendix B

Associate Editor

Board Member: 1

Comments to Author:

I have obtained two thoughtful and thorough reviews of the manuscript. Both reviewers see significant value in this manuscript (though one reviewer is considerably more enthusiastic than the other) and I fully agree in that assessment. I therefore encourage the authors to revise the manuscript in the light of both reviews. Further, I would like to add two comments: (i) Figure 1 could be improved in all aspects, from light and contrast to focus, perhaps by generating composite images from z-stacks; (ii) the entire introduction and discussion are eusocial hymenoptera focused. But there are other eusocial insects out there for which dietary information is available (e.g. termites), and there are non-eusocial insects out there which are nonetheless polyphenic and both diet quality and quantity effects have been assessed (e.g. dung beetles). Broadening the discussion accordingly may allow the authors to place their results into a broader context to further explore the distinctness (or lack thereof) of caste determination in honey bees.

Regarding Fig 1: We have remade Fig 1 so that it is a line drawing instead of a photograph. This has made the measurements clearer and saved space, which allowed us to also increase the size of the body parts relative to the size of the figure panel.

Regarding broadening the Discussion: We revised the final paragraph of the discussion to put our results in a broader context. We have added citations for dung beetles and termites in addition to mentioning nutritional regulation of reproduction in meerkats.

Referee 1

This manuscript reports much needed work that examined the effect of food quantity on caste determination in honey bees. That royal jelly is the magic food enabling queen development has captured the imagination for centuries and is similarly dated. The belief that it contains one single, magic ingredient that effects the development switch is unwarranted.

Comments

Line 71. THE FACT THAT Elevated JH ALSO induces queen develop in honey bees, WHICH HAVE MORE COMPLEX SOCIALITY, suggests a conserved mechanism IN WHICH DIET WORKS THROUGH JH TO PRODUCE INDIVIDUALS WITH HIGHER REPRODUCTIVE POTENTIAL.

Or some other way of expressing explicitly what conserved you are thinking of.

The sentence now reads:

“The fact that elevated juvenile hormone also induces queen development in honey bees (31, 32), suggests a conserved mechanism in which reproductive status is regulated by diet and mediated by juvenile hormone.”

Royal jelly is used in all treatments – so the all the other components in it are still there though in less amounts. Is this a problem?

For example, might there be sufficient royalactin to maintain pluripotency, in most treatments? See Nature Communications 2018 Wan et al. Honey bee royalactin unlocks conserved pluripotency in mammals.

The fact that royalactin can help mouse stem cell maintain pluripotency is indeed intriguing. Wan et al. discovered that there was a mouse protein that has similar functional domains as Royalactin, suggesting that is why Royalactin can perform this function in mammals. They indicate that Royalactin maintains pluripotency in mammals by altering chromosomal structure-increasing access to specific regions of the genome. To our knowledge, it has not been determined that Royalactin alters chromosomal structure in honey bees, or that it is involved in maintaining cell pluripotency in honey bees. It is known that changes in methylation accompany caste determination, and it is possible that Royalactin mediates these methylation differences, but we don't think that link has been explicitly tested. In our opinion, it is not clear what is "conserved" about Royalactin and its similar mammalian protein. They both induce similar outcomes in mice cells, but that could also be due to convergent evolution. And, I am not sure that the maintenance of pluripotency would be a factor in caste determination. Do queen-destined larvae have cells with greater pluripotency? We would be interested to know what Referee 1 thinks of the matter.

As regarding the initial question: Does the fact that Royalactin is present in all treatments cause as sufficient level of activity to make queen traits? The reasoning being that even in the lowest quality diets, there is enough Royalactin to induce queen-like traits. If that had been the case, then we would have expected more queen-like traits in diets with higher percentages of Royal Jelly but smaller overall quantities. We did not get that result.

243. It is amazing that ad libitum individuals do as well as commercially reared queens. This is such an important finding. Love it.

Thank you.

251. In addition to cell size, they change the orientation. This is apparently related to higher viscosity of RJ structure and viscosity (something in Insectes Soc. Recently). This may be another way quantity is controlled in the hive setting. Another mystery here.

We have added a citation that work in the Discussion (Line 315-316):

The full reference is:

Buttstedt A, Muresan CI, Lilie H, Hause G, Ihling CH, Schulze SH, et al. How Honeybees Defy Gravity with Royal Jelly to Raise Queens. *Curr Biol.* 2018;28(7):1095-100 e3.

278. Clarify that TOR is in the IIS pathway and EGFR is not? TOR is not independent of JH. Mutti (31) title says IRS and TOR pathways act via JH. In addition, add that JH was artificially applied in Mutti's case.

We have edited that passage to say the following (lines 275-277):

"In honey bees, the insulin pathways links nutritional status to juvenile hormone production (31, 47-49), with the involvement TOR (31, 49, 50). Ostensibly, these pathways can either trigger juvenile hormone release from the corpora allata (32) or be triggered by juvenile hormone, as observed during external application of the hormone (31)."

329. Yes a loss of reproductive function but so much more is involved, such as numerous other features associated brood care and foraging.

We have added the follow sentence to that section (now line 349):

“In honey bees, the loss of reproductive potential in workers coincides with the development of multiple morphological and behavioral traits that support hive fitness.”

336. I disagree with that the study suggest that larvae are predetermined queens but for starvation and other abuse. Any (most) organisms have within them the information to develop robustly under that best of conditions – but those conditions rarely exist so it doesn't mean the most robust form is the default. Queen determination is a bifurcating developmental system in which nutritional cues are manipulated and assessed through multiple pathways to respond to conditions they experience. Both castes are highly specialized outcomes.

We have extensively revised the final paragraph of the discussion and removed the statement that larvae are predetermined queens. Instead, we focus on how food restriction is a possible mechanism for induction of the worker caste.

OTHER

The Bowsher lab has also published a paper on honey bee development and the geometric model of nutritional needs. I would love to see these findings integrated here. Should we expect this model to work when nutrition and its developmental outcomes are not unitary?

We have added a citation to that work (Helm et al. 2017) in the Discussion (citation 54). The discussion has been significantly revised and the geometric framework is now discussed here (Lines 328-332):

“Diet quality is important for juvenile growth, development, and survival (34, 35, 54). Growth is the result of not one but many interacting nutrients. When protein and sugar content are altered under controlled diet quantities, as done under a geometric framework for nutrition, growth rate, development and survival are affected (54). We observed a similar result of high mortality on low protein diets.”

Misc .

Line 45, feed=fed

Line 220, diet quality. If you mean quality, than I am confused by this sentence.

And finally, are the percentages really significant to 2 decimal places?

Line 248. and instead of but

302. A result

313. Bee larvae (of) on – delete of.

We have made all of the above changes.

Referee: 2

Comments to the Author(s)

Slater et al. challenge conventional wisdom about how caste is determined in honey bees. They do this through *in vitro* rearing of honey bees under different diet regimes varying in quality and quantity in a fully factorial design, characterizing the resulting adults as queens, workers, or intercastes based on external morphology. Caste determination is an important feature of eusocial insects and honey bees have played a major role in understanding social behavior, and thus the study has relevance to the field. However, I have a number of concerns about the methods and interpretation of the results that prevent me from accepting their conclusions at face value.

The diet treatments used are based upon diets previously shown to induce worker development in honey bees (the lowest diet quantity and the medium-protein medium-carbohydrate diets are cited from another study as worker-inducing). The other diet treatments involved larger quantities relative to this worker reference diet, as well as mixes of royal jelly with glucose and fructose to vary “quality”. I would like to have seen a diet treatment of entirely royal jelly, or at least diets that were closer in “quality” to pure royal jelly. That aside, the diets they did use led to some surprising effects. First, the lowest quantity diet produced more intercastes than workers. Table 3 doesn’t break down the numbers of individuals in each cluster by quality (these data need to be available in a supplemental table!), but seeing 19 intercastes and 10 workers eclose from larvae reared on the lowest quantity diet is surprising, and to me indicates that quality MAY matter, as the variation in caste must be coming from somewhere. I see no discussion about why, even at the lowest quantity tested, there is so much variation in phenotype. Is *in vitro* rearing wildly different between labs such that the same diet in one lab leads to very different results in another? I worry about the relevance of the findings with respect to natural caste determination in honey bees (as opposed to lab reared bees).

With regards to a 100% Royal Jelly treatment: Many researchers have tried to rear larvae *in vitro* on only Royal Jelly and have fail. We have also attempted to do so and were not successful. In line with these attempts, our high protein diet had a high mortality, and this shows a pure royal jelly diet is not possible, especially if we want high survival. This book (which we have cited in the manuscript) goes into great detail about the failures of *in vitro* rearing with pure royal jelly: Singh, P. (1977). Artificial diets for insects, mites, and spiders (No. 638.5 S5).

With regard to Table 3: We have added a new table, Table S2, which breaks down the number of individuals per cluster for each diet quality. We refer to this table in the Results (Line 209-210):

“In contrast, the diet quality treatments of *in vitro* reared bees were more evenly distributed across clusters, with 22-39% clustering with queens for the high and medium protein diets (Table S2).”

Regarding the amount of variation found in this study:

In vitro rearing protocols almost always result in some intercaste individuals. The cause of this variation is not clear because of the challenge of standardizing between studies. Each batch of Royal Jelly is different so the macronutrient content has to be measured (as we did) to make a diet consistent across studies. Also, there is not a standard protocol for how much the larvae are fed each day, so that varies between studies (and was one of the motivations for our study).

The motivation for choosing the medium protein, medium sugar diet as the reference was the fact that those nutrient proportions are the most commonly used in *in vitro* studies, and Aupinel (2005) found 160ul produced the most workers compared to other qualities or quantities. However, De Souza et al 2015 found bees reared on this diet produced a spectrum of workers to queens, so this variation was not unexpected. It may have been the case that diet quantities below 160ul would have produced 100% workers, but we did not try that because likely would have had high mortality.

I also struggle to separate quality from quantity in their results and interpretation. The authors suggest that there is no “biological active substance” in royal jelly that makes a queen. I don’t think this conclusion can be made with the present data. There MAY be some component of royal jelly that is necessary, above some threshold, in order to develop queen traits. This threshold could be reached by a lower quantity of a high “quality” diet (pure royal jelly would have more than diluted royal jelly) OR by a larger quantity of a low “quality” diet. In other words, “quantity has a quality all its own”. There may be a compensatory effect of quantity that makes up for a lack in quality, but that doesn’t mean that diet quality doesn’t have an effect with natural diets under natural conditions. Further, the range in diet quantity tested appears, at least superficially, a lot greater than the range in quality of diets (e.g., 2.3-fold variation in quantity, but only 1.5-fold variation in protein content). This, combined with the fact that a queen-like diet wasn’t tested (pure royal jelly), makes me wonder if an effect of quality would have been seen with a more realistic (fully worker to fully queen) range of diet qualities.

With regard to separating the effects of quantity and quality: Our goal for this study was to measure the effects of quantity vs quality by changing them factorially. We determined that poorer quality diets can still produce queen-like traits when fed in excess. But we also have evidence for the reverse: high quality diets do not make queen-like traits if fed in small amounts. We also did not find an interaction between quantity and quality. We agree with the reviewer that there is likely a threshold of a certain amount of nutrition that is required for queens, such that poor quality diets can still produce queen-like traits in large amounts. However, based on our results, that effect is not tied to either protein or carbohydrate proportion.

Regarding natural conditions: The reviewer points out that *in vitro* effects may not mimic natural conditions. We agree. *In vitro* rearing produced a lot of intercastes individuals, and intercastes are very rarely, if ever, found in hives. Because the nurse bees are the proximate regulators of larval caste determination through feeding behavior, we guess that they prevent the development of intercastes. We chose to do this experiment *in vitro* because it is technically very challenging to determine the amount of food each larva is fed over its development in a hive setting. We hope that this *in vitro* experiment will inspire work that will test these ideas in the hive.

Regarding the range of diets: The range of diet quality was more constrained than diet quantity because changing the protein variation beyond 1.5-fold push the thresholds of survival. Even with the 1.5-fold range, we had very high mortality in the low protein diets. As we have described above, larva die on a diet of pure royal jelly *in vitro*, so upper limit of royal jelly content was also a constraint.

To address these concerns, we have significantly edited the Discussion section. These edits tone down the rhetoric and explain in more details the limitations of this work, and its context within other studies on caste determination.

I understand that there are limitations in what can be tested, and the authors mention that casein and other more artificial diets cannot be used for larval rearing of honey bees. However, using royal jelly as a protein source comes with a range of complications, some of which could at least be taken into consideration in the discussion of this manuscript. Other authors have used heat to denature the proteins in royal jelly and have seen effects on caste determination (inability to rear queens). Can the authors speak to these previous findings? Did the authors experiment with trying to “deactivate” any putative “biological active” substances in the royal jelly? I struggle to view royal jelly as just a source of protein, and reducing its content to a percent protein, carbohydrate, and water seems unrealistic.

We have significantly edited the discuss which includes the section where the limitations of not having an artificial diet are discussed. These edits provide more detail on the limitations of an *in vitro* set up.

Kamakura 2011 heated royal jelly for 30 days at 40C and found very small ovary size and worker like characteristics. The protein was completely degraded and they did not look at the integrity of other macronutrients besides protein. We think heating royal jelly at this temperature and for that long may have had effects on the other macronutrients, reducing the overall quality of the diet to a level below what any hive-reared bee would receive. He did not report survival. We think the effect of on caste cannot be entirely ascribed to the denatured royal jelly.

In addition, I have a number of minor questions, comments, and suggestions for improvements:

Line 34: “caste” should be “castes”

We have made this change

Line 40 and 234: not all eusocial insects show environmentally-determined castes (for example, some stingless bees have genetic caste determination)

We have changed line 40 to read:

“In bees, queens and workers have analogous genotypes; yet, these similar genomes produce distinct queen-worker phenotypes. Caste determination cues vary by species, but nutrition drives queen development in many social Hymenoptera (1, 2).”

We have changed line 236 to read:

“In many bee species, environmental cues drive female larvae into irreversible queen or worker developmental pathways (10).”

Line 55: “gland” should be “glands”, or maybe “glandular secretions”?

We have made this change to “glands”

Line 59: no comma before “all”

We have made this change

Line 62: how much diet do workers get? 1.5 g for queens, would be nice to compare with workers, and also compare the diets used in this study (given in volumes) to these numbers

The reference that provided the 1.5 g for queen larva (Haydak 1970) did not provide a number for workers. The only reference we could find that measured larval food amounts for workers was Allsop et al 2003 and that work was in the cape honey bee. The amounts reported were 4.4 mg for workers and 7.4 mg for queens. These numbers for the cape honey bee are well below what we would expect for a European honey bee based on Haydak (1970). So, we were not able to find a value for the total amount fed to worker honey bees.

Line 69: add “an” before “elongated”

We have made this change

Line 71: add “also” before “induces”

We have made this change

Lines 105-107: Discussion of a feeding stage makes me wonder- can larvae prolong their feeding stage? Could larvae receiving lower quantity/quality diets extend the time between instars or prior to pupation? In this way, could they make up for any deficits due to lower quantity/quality?

This is a good question. We don't think larvae can prolong their feeding stage independent of nurse provisioning. We suspect that the length of the fifth instar, which is the stage with the greatest increase in size, is regulated by nurse bees. We have evidence from solitary bees that they will feed until their food provision is gone, and then they initiate metamorphosis (Helm et al 2017). If we feed them excess food, they will continue to feed and will grow larger than the control. We have some preliminary data showing that honey bees may be similar—that removing food initiates metamorphosis. This would mean that nurse bees not only regulate size through feeding, but also can initiate metamorphosis by restricting food. This also means that nurse bees have the potential to prolong the fifth instar by providing excess food. But, these data are preliminary.

Helm, B., Rinehart, J., Yocum, G., Greenlee, K., J. H. Bowsher. 2017. Metamorphosis is induced by food absence rather than a critical weight in the solitary bee, *Osmia lignaria*. *Proceedings of the National Academy of Sciences*, 114(41), 10924-10929.

Line 114: “can be mixed into...” better written as “have been previously used to alter carbohydrate content of in vitro diets”

We have made this change

Lines 120-121: Was the same stock of royal jelly used for all treatments? Are the protein, carbohydrate, and water contents of this royal jelly similar to others used? How consistent are the macronutrients of royal jelly?

The same stock of royal jelly was used for all treatments. Commercial royal jelly has a range of macronutrient contents, as do samples that have been analyzed from hives.

Morphometrics section: For confirmatory purposes, it would have been nice if variation in ovariole number and spermathecal size were examined in a subset of individuals.

We agree. Because we decided to freeze the bees upon emergence, we were not able to go back and collect that data.

Line 160: “hierarchal” should be “hierarchical”

We have made this change

Line 179: remove “The”

We have made this change

Line 180: seven morphological measurements mentioned, but only see 6 in figure (eventually realized the 7th was adult weight, but not clear until later)

We have changed this line to read:

“Principal Component Analysis (PCA) was used to distinguish reference queens from reference workers based on six morphological measurements and body mass (Fig. 1).”

Fig. 2- sample sizes should be indicated in legend; difficult to discern all groups with colors used, maybe rather than a ROYGBIV color scheme a gradient of one color for quantity would be more clear? In addition, it would be nice to see a similar figure which colors individuals by quality rather than quantity which would presumably show a lack of grouping by quality?

We added sample sizes to the legend.

We attempted to revised Fig 2 using a viridis gradient color scheme, but it is very difficult to make an eleven-tone range that provides a visible difference between tones while also avoids very light tones (white-ish or light yellow) that blend into the background. Instead, we have revised Fig 2 to be a three panel graph which is divided between high, medium and low diet quantities. This allows visual separation between the groups.

We also added a new supplemental figure (Fig S2) that color codes individuals by quality. It is a three panel graph separated by protein content (high, medium and low).

Fig. 2 and 3- legends are not consistent in their ordering, which threw me initially

Line 192: space in “PC1and”

We added the space. We have revised Fig 2 to group panel by quantity.

Fig. 3- very difficult to parse the individuals and colors; can leaves of the same color be combined into larger color blocks? Again, colors are difficult to discern with respect to quantity groupings

The leaves come up against each other without a gap in between. So, adjacent leaves can make a continuous block. In order to make larger blocks, we would have to collapse nodes into unresolved polytomies (perhaps 4 layers of nodes?), which would reduce the amount of information in the figure. The colors in this figure match those in Fig 2. We have not been able to determine a way out of the eleven tones that are required.

Table 3- at LEAST in a supplemental table, I really desperately want to see a breakdown of queen/worker/intercaste clusters by quality. Also, why not label clusters as queen/worker/intercaste as opposed to numbers?

We have added a Table S2 that clusters by quality.

We used numbers to designate clusters because we wanted to emphasize that we did not assign workers and queens to clusters at the start of the clustering analysis. Instead, the software assigned individuals to clusters without knowing they were queen or workers. It happened that all the reference queens clustered together and all the reference workers clustered together. That outcome was not surprising, but it would have been possible for a reference queen or worker to end up in the cluster with intercastes.

Lines 207-209: While the ad libitum treatment produced all queens, I find 58% from the highest quantity diet lower than expected if quantity is so important. One big difference, if I am interpreting the methods correctly, is that the ad libitum group received more diet throughout larval development, while the other diet treatments only varied in the final stages of development. Wouldn't this indicate that timing is also important, in addition to quantity? Could the authors speak to this?

The reviewer is correct on the methods. The ad libitum treatment was fed more throughout development. We agree that timing is probably important. The third instar is considered a critical period in caste determination. We were able to induce queen-like traits by excessively feeding on the sixth day, but the reviewer is correct that this method produced 58% of queen-like individuals and no more. This indicated that early excess feeding is a component of queen determination. However, it is hard for us to say more to that point based on our experiment. During the course of the experiment, we did not have any larvae finish all their food before they were fed the next day. There was always a bit of excess food left. So, although the factorial treatment received less per day than the ad libitum treatment, they did not consume all their food until the sixth and final day.

Line 220: is "quality" meant to be "quantity" here? Either an important typo or I'm confused

We have changed it to quantity.

Line 226: I don't like a sentence that starts with "And,"

We have deleted “And.”

Line 233-234: Division of labor is not really why I view eusocial insects as a good example of phenotypic plasticity; strange phrasing at the very least

We have removed “division of labor.”

Lines 236-237: which studies have “fully characterized” the cues that drive caste divergence? Citations?

We have added citations.

Line 245: 19/33 is technically most, sure, but there are plenty of the high quantity individuals that did NOT cluster with commercially-reared queens, perhaps indicating that the range of quantities/qualities used is not representative of the full range of relevant diets to compare queen and worker caste determination

See response to the main comments above where we discuss the range of diets.

Lines 256-257: do ovariole number and spermathecal size measurements really have to be done immediately after collection? Why couldn't individuals be frozen first? And as far as mortality, I believe the authors killed them anyway, so dissection should have been possible

We froze the bee upon eclosion so that we could later measure wet weight and the morphological characteristics. Upon freezing, ice crystals destroy the integrity of soft tissues. When thawed, tissues are more liquid and fragile than when fresh. Because ovarioles and spermatheca are already relatively soft and fragile tissues, measuring them after freezing would have been possible.

Line 271: “Many social hymenopterans use diet quantity to regulate caste determination”. I would argue that obviously honey bees do this as well- queen honey bees are fed much more...

Yes. We also argue that honey bees using quantity to regulate caste.

Line 277: “links” should be “link”

We have made this change.

Line 278: missing word “of” before TOR

We have made this change.

Line 307: MRJP not MJRP

We have made this change.

Line 309: “may not differ” is strange phrasing; “do not always differ”?

We have made this change.

Line 310: are the authors referring to all proteins, or specifically MRJP1 and 3?

The line now reads:

Royal jelly proteins are thought to regulate caste, but royal jelly and worker jelly do not always differ in the proportion of Major Royal Jelly Proteins.

Line 313: extra “of”

That line has been removed during editing.

Line 328: respect “to” sociality, not “of”

We have made this change.

Line 346: “Ours” should be “Our”

We have made this change.

Line 346: I disagree that diet quantity is “unacknowledged” with respect to honey bee caste determination- anyone who has ever seen a queen cell will attest to the fact that certainly, queens receive a lot more food than workers

We have edited this section to state that diet quantity has not been explicitly tested as a mechanism of caste determination.

Appendix C

Reviewer(s)' Comments to Author:

Referee: 2

Comments to the Author(s).

I appreciate the authors' consideration of my previous feedback and feel the manuscript has been improved to a state worthy of publication. The finding that diet quantity, rather than quality, influences caste determination in honey bees is an important one that will influence the field of sociobiology.

I have just a few additional minor comments/changes to suggest:

Lines 28-30: "We found that larvae fed the largest quantities of diet were indistinguishable from hive reared queens, independent of the proportion of protein and carbohydrate in the diet"

- Not all larvae fed the largest quantities were indistinguishable from queens, right? Just a subset? Unless "largest quantities" refers to ad lib, which I think is misleading because the abstract refers to 8 different quantities, and there are 8 plus ad lib.

We have revised the previous lines to include information on the ad lib treatment, and also qualified the lines mentioned by the reviewer. The section now reads:

"Larvae were reared *in vitro* on nine diets varying in amount of royal jelly and sugars, which were fed to larvae in eight different quantities. For the middle diet, an *ad libitum* quantity treatment was included. Once adults eclosed, the queenliness was determined using Principal Component Analysis (PCA) on seven morphological measurements. We found that larvae fed an *ad libitum* quantity of diet were indistinguishable from commercially reared queens, and that queenliness was independent of the proportion of protein and carbohydrate in the diet."

Line 42: "queen-worker phenotypes"

- Queen and worker phenotypes?

We have made this change.

Line 63: Is 1.5g the amount extra, or the total amount fed to the queen, not all of which is consumed? Not clear.

We have removed the reference to the specific quantity of 1.5 g because the reference (Haydak 1970, reference #3) did not provide a worker quantity of diet for comparison. The sentence now reads: Nurse bees provide queen-destined larvae with an excess of diet (3, 6, 8, 21).

Line 73: should not be a comma after (31, 32) references

We have made this change.

Line 94: "eight diet qualities and nine quantities" vs lines 25-26 "Larvae were reared in vitro on nine diets varying in amount of royal jelly and sugars, which were fed to larvae in eight different quantities."

We had a factorial design of nine diet qualities and eight quantities. There was an additional treatment group that was an ad libitum quantity of the medium-protein, medium-carb diet. We have changed both lines 25-26 and Line 94 to correct the error and clarify the experimental design. Line 94 now reads: "Larvae were fed according to treatment in a factorial design of nine diet qualities and eight quantities with an additional *ad libitum* treatment for the medium diet, as described in the following sections."

Methods: I would add a sentence clarifying that the same royal jelly stock was used for all diets for the duration of the experiment

We have added that sentence at Line 125-126.

Line 132: “pupations” should be “pupation”

We have made this change.

Line 150: “intercastes” should be “intercaste”

We have made this change.

Line 211: “with 22-39% clustering with queens for the high and medium protein diets (Table S2).”
- For high and medium protein diets it appears the range is 19-39%, not 22-39%

We have changed to 19-39%.

Lines 217-218: “ad libitum treatment were not included in the model because the treatment was not factorial for QUANTITY”

- I assume the authors mean quality, not quantity here?

We have changed the line to read:

“The individuals from the *ad libitum* treatment were not included in the model because the treatment was not part of the factorial design.”

Depending on the interpretation, the *ad libitum* was not factorial for either quantity or quality because that treatment was only represented by one of the diet qualities and included a quantity that was not used for any of the other diets.

Line 248: “influences” should be “influenced”

We have made this change.

Line 249: “first to directly test the role of diet quantity [on honey bee queen development]”

- I recommend adding the qualifier for clarity

We have made this change.

Line 255: maybe more clear as “caste is empirically determined based upon”

We have made this change.

Line 296: if MRJP-1 and MRJP-3 influence queen development in the absence of controlling quantity, does this mean these proteins may influence larvae to consume more food, thus leading to queenliness? I wonder if the authors could speculate about the role these proteins play in consumption.

Carbohydrates are known to increase consumption in insects, which we discuss in lines 350-354. We do not know of any studies that link MRJPs to increased consumption. If MRJPs did increase consumption, then that would possibly explain how quality and quantity could be related. However, we do not know of

any evidence to support the idea.

Line 299: might as well use the MRJP-1 and MRJP-3 abbreviations (but throughout this paragraph the abbreviation is incorrectly MJRP rather than MRJP)

We have made this change.

Line 302: Royalactin is capitalized here but not elsewhere, and the abbreviation has already been given twice so no need to do that again

We have made this change.

Line 335: “tested” should be “testing”

We have made this change.

There are typos and inconsistent formatting in the references; please check these carefully.

We have gone through the references and fixed formatting issues.